# Synthesis of Novel Benzoxazines Containing Sulfur and Their Application in Rubber Compounds

**DOI:** 10.3390/polym13081262

**Published:** 2021-04-13

**Authors:** Acerina Trejo-Machin, João Paulo Cosas Fernandes, Laura Puchot, Suzanne Balko, Marcel Wirtz, Marc Weydert, Pierre Verge

**Affiliations:** 1Luxembourg Institute of Science and Technology, L-4362 Esch-sur-Alzette, Luxembourg; acerina.trejomachin@list.lu (A.T.-M.); joao.cosas@list.lu (J.P.C.F.); laura.puchot@list.lu (L.P.); 2Department of Physics and Materials Science, University of Luxembourg, L-4365 Esch-sur-Alzette, Luxembourg; 3Goodyear Innovation Center Luxembourg, L-7750 Colmar-Berg, Luxembourg; suzanne_balko@goodyear.com (S.B.); marcel_wirtz@goodyear.com (M.W.); marc_weydert@goodyear.com (M.W.)

**Keywords:** benzoxazines, rubber, reinforcing resins, polyisoprene, disulfide

## Abstract

This work reports the synthesis and successful use of novel benzoxazines as reinforcing resins in polyisoprene rubber compounds. For this purpose, three new dibenzoxazines containing one (4DTP-fa) or two heteroatoms of sulfur (3DPDS-fa and 4DPDS-fa) were synthesized following a Mannich condensation reaction. The structural features of each benzoxazine precursor were characterized by ^1^H and ^13^C nuclear magnetic resonance (NMR), Fourier transform infrared (FTIR) and Raman. The new precursors showed well suited reactivity as characterized by differential scanning calorimetry (DSC) and rheology and were incorporated in rubber compounds. After the mixing, the curing profiles, morphologies and mechanical properties of the materials were tested. These results show that the structural feature of each isomer was significantly affecting its behavior during the curing of the rubber compounds. Among the tested benzoxazines, 3DPDS-fa exhibited the best ability to reinforce the rubber compound even compared to common phenolic resin. These results prove the feasibility to reinforce rubber compounds with benzoxazine resins as a possible alternative to replace conventional phenolic resins. This paper provides the first guide to use benzoxazines as reinforcing resins for rubber applications, based on their curing kinetics.

## 1. Introduction

One of the most important fields of study in material science concerns the development of rubber formulations [1]. These materials are very convenient due to their elasticity and damping properties once crosslinked [2]. However, rubber compounds need to be reinforced to fit the specifications of mechanical properties required for many applications [3,4]. For this purpose, inorganic fillers or polymeric resins are commonly employed. Among the polymeric resins, novolac-type phenolic resins (PR) are the most employed [5,6]. These PR are formaldehyde pre-condensates from phenol or resorcinol [7]. In most of the cases, they are incorporated into the rubber with an *in-situ* crosslinker such as hexamethylenetetramine and react while the rubber is curing [8,9]. Even though a good performance is observed, several shortcomings are arising from their use such as the employment of a methylene donor to allow the resin to crosslink. The development of suitable alternatives has become desirable [10,11]. 

Among the different alternatives, polybenzoxazines (polyBz) have emerged as promising substitutes for phenolic resins. Benzoxazine resins (Bz) are readily synthesized in an one-step reaction through a Mannich-like condensation of phenolic compounds, formaldehyde, and primary amines [12]. Then, they are subjected to being thermally triggered, cationic ring-opening polymerization (ROP). This polymerization is auto-catalyzed, and it is commonly activated in a temperature range between 180 and 250 °C [13]. These mono-component resins do not release phenols, or formaldehyde during the polymerization step [12]. Moreover, polyBz show excellent mechanical properties, enabling their use as reinforcing resins for rubber applications [12,14]. 

Up to now, the combination of rubber and benzoxazines has been mainly focused on the improvement of the toughness of polybenzoxazine matrixes. One of the most studied strategies is the incorporation of reactive liquid rubbers into Bz to lower the curing temperature while the thermal, tribological, and/or mechanical properties were improved [15,16,17,18,19]. Additionally, benzoxazines have been employed to functionalize rubbers. For example, Bz moieties were anchored onto the backbone of polybutadiene rubber by click chemistry leading to a self-curable rubber [20]. Thio-ene click reactions were also carried out to develop a Bz-functionalized poly(styrene-butadiene) rubber that exhibits improved mechanical properties [21]. Another example focused on the preparation of a polysulfide rubber chain bridged with polyBz by the ring-opening addition of a thiol-capped rubber to obtain a material with outstanding thermal stability and excellent flexibility [22]. 

Regarding the application of polybenzoxazines to reinforce rubber compounds, just a limited amount of patents can be found [23,24,25,26,27,28]. In 2007, the use of two di-benzoxazines (diBz) commercially available was patented, one from bisphenol A (BA) and aniline, and another from phenol and 4,4′-diaminodiphenylmethane (ddm) in rubber compounds [23]. The addition of the diBzs improved the elasticity and the breaking strength. Recently, in 2020, a patent was granted concerning the use of linear polymers with benzoxazines groups in the backbone known as main chain benzoxazines from BA and ddm, terminated with monoamines to reinforce filled rubber compounds [28]. The literature concerning this topic is not rich despite the potential scope of this field. Two main reasons could explain this lack of studies. The first one is the disparities on the curing kinetics of benzoxazines and rubber. The curing of rubber by sulfur, so-called vulcanization, is commonly carried out at temperatures between 150 and 170 °C in a relatively short period of time [2] while benzoxazine ROP requires high temperatures and comparably long reaction times to complete polymerization [29]. Secondly, interactions between the rubber curing additives and benzoxazines could occur when they are cured together. Indeed, Liu et al. reported a new reaction between sulfur and benzoxazine to form a copolymer at temperatures above 159 °C known as sulfur radical transfer and coupling reaction (SRTC) [30]. Moreover, benzoxazines containing double bonds can react with elemental sulfur to produce copolymers under curing conditions. In these cases, inverse vulcanization and ring-opening polymerization take place concomitantly to produce a wide variety of materials [31,32,33,34]. Finally, elemental sulfur has been reported to trigger the formation of *in-situ* initiators that reduce the polymerization temperature of benzoxazines [35]. 

This work reports the synthesis and successful use of novel benzoxazines as reinforcing resins in polyisoprene rubber compounds. The chemical structures of the monomers, which are represented in Scheme 1, and their reactivity appeared to be a major parameter to consider in the elaboration of the rubber compounds. The polymerization kinetics were investigated in detail in the absence and presence of sulfur and rubber curing additives. Once the behavior of the monomers was understood, they were incorporated in polyisoprene formulations and tested as reinforcing agents. This paper relates the curing behavior, morphology characterization, and mechanical properties of the rubber compounds containing benzoxazines and its potential use to replace phenolic resins.

## 2. Materials and Methods

### 2.1. Materials

4,4′-thiodiphenol (99%, 4TDP), furfurylamine (≥99%, fa), bisphenol A (≥99%, BA), and paraformaldehyde (95%, PFA) were purchased from Sigma-Aldrich. 3,3’-dihydroxydiphenyl disulfide (95%, 3DPDS) was purchased from abcr GmbH, and 4,4′-dihydroxydiphenyl disulfide (98%, 4DPDS) was purchased from TCI Europe. Phenolic resin (>99.5%, PR), with product name DUREZ 31459 was purchased from SBHPP (Sumitomo Bakelite Co., Ltd.). Hexamethylenetetramine (HMT) was supplied by Ineos Paraform. Toluene (≥99.5%), chloroform (CHCl_3_), chloroform-d (CDCl_3_) and dimethylformamide (DMF) were supplied from Sigma-Aldrich. All chemicals were used as received without any further purification.

### 2.2. Methods

#### 2.2.1. Nuclear Magnetic Resonance (NMR)

Nuclear magnetic resonance (NMR) spectra were recorded using an AVANCE III HD Bruker spectrometer operating at 600 MHz and equipped with a 5 mm Broadband Observe (BBO) probe. The samples were dissolved in deuterated chloroform (CDCl_3_) and the spectra were referenced relative to tetramethylsilane (TMS). Assignments were performed using a combination of ^1^H, ^13^C, homonuclear correlation spectroscopy (COSY), heteronuclear single quantum coherence (HSQC), and heteronuclear multiple bond correlation (HMBC) spectra.

#### 2.2.2. Fourier Transform Infrared Spectroscopy (FTIR)

Fourier transform infrared spectroscopy (FTIR) was conducted on a Bruker TENSOR 27 (Ettlingen, Germany) in transmission mode. The background and sample spectra were recorded at 4 cm^−1^ spectral resolution across the 4000–400 cm^−1^ range.

#### 2.2.3. Elemental Analysis

The elemental analysis measurements, which provides the determination of carbon, hydrogen, nitrogen and sulfur (CHNS), were performed on a Vario MACRO cube CHNS/O from Elementar France SARL. Samples are inserted in an oxygen enriched furnace at 1150 °C where a combustion process converts carbon to carbon dioxide; hydrogen to water; nitrogen to nitrogen gas/oxides of nitrogen and sulfur to sulfur dioxide. The combustion products are swept out of the combustion chamber by inert carrier gas (helium, 600 mL per minute) and passed over heated (850 °C) high purity copper. The separation of the measuring components takes place as follows: N_2_ is not adsorbed in the adsorption columns and is the first measuring component to enter directly in the thermal conductivity detector. The other components are adsorbed in their respective adsorption column. Each of these columns is then separately heated to the corresponding desorption temperature (T_desorpt._) in order to release the components in the following order: CO_2_ (T_desorpt._ 240 °C), H_2_O (T_desorpt._ 150 °C) and SO_2_ (T_desorpt._ 230 °C). After desorption, each component is transported by the carrier gas flow into the measuring cell of a thermal conductivity detector (TCD).

#### 2.2.4. Raman Spectroscopy

Raman spectra were recorded in a back-scattering geometry with a Renishaw inVia ReflexRaman Microscope using the 785 nm line of High Power Near Infrared Diode Laser, 3 accumulations of 10 s were used at power 24.3 mW. An X50 long working distance objective was used to focus the laser beam on a sample surface.

#### 2.2.5. Differential Scanning Calorimetry (DSC)

Differential Scanning Calorimetry (DSC) thermograms were recorded using a Mettler Toledo DSC3+ apparatus operating at inert atmosphere (nitrogen) with a linear heating ramp from 20 to 250 °C at 10 °C·min^−1^ rate.

#### 2.2.6. Thermo-Gravimetric Analysis (TGA)

Thermo-Gravimetric Analysis (TGA) measurements were performed using a Netzsch TG 409 PC Luxx device operating under nitrogen with a heating ramp of 10 °C·min^−1^ up to 800 °C.

#### 2.2.7. Rheological Measurements

Rheo-kinetic measurements were performed using an Anton Paar Physica Modular Compact Rheometer (MCR) 302 rheometer equipped with a CTD 450 temperature control device with a disposable aluminum plate-plate (diameter: 25 mm, measure gap: 0.5 mm) geometry. The polymerization measurements were recorded in the oscillation mode with linear strain amplitude from 1 to 0.1% and a frequency of 1 Hz. The test is performed following a heating ramp of 20 °C·min^−1^ from 50 °C to 150 °C followed by an isothermal measurement at 150 °C. Rheology temperature sweep curves were performed on the same device on cured rectangular bars in torsion mode from room temperature to 300 °C under constant deformation of 0.1% and a frequency of 1 Hz.

#### 2.2.8. Preparation of Polybenzoxazines

Benzoxazine monomers were placed in hollow Teflon^®^ molds and dried under vacuum for 2 h at 100 °C to remove traces of solvent or water. After that, the molds were transferred to an air-circulating oven at 170 °C for one hour for the first curing step. This step was followed by a post-curing for 1 h at 190 °C and then 1 h at 210 °C.

#### 2.2.9. Solubility Tests

Solubility was assessed in dimethylformamide (DMF). 0.1 g (±0.01g) of benzoxazine monomers without and with 30 %wt of sulfur were cured at 150 °C for 1 h. After that, they were immersed in 5 mL of DMF for 3 h. 

#### 2.2.10. Rubber Compounding

A Thermo Scientific HAAKE PolyLab QC internal mixer was used to perform the compounding. The internal mixer having a free volume of 85 cc was operated using two cam-type rotors and keeping a constant fill factor of 0.75. The mixing process was performed in two steps. The initial temperature was set to 70 °C for the first process, also referred to as non-productive (NP) step, and to 60 °C for the final one, also known as productive step (PD). A formulation containing polyisoprene was mixed with three different amounts of each benzoxazine as shown in Table 1. All the values are indicated in weight percentage as well as in phr (parts per hundred rubber), which is commonly used in the rubber industry. Additionally, the mixing procedure is detailed in the Section A.1 (Table A1).

#### 2.2.11. Moving Die Rheometer (MDR)

An Alpha Technologies moving die rheometer (MDR) 2000 was used to measure the cure kinetics of rubber compounds. The MDR was preheated at 150 °C for approximately 30 min. After that the test was performed at 150 °C for 80 min with an oscillation of amplitude of 0.5° (~7% strain) and a frequency of 1.667 Hz. Optimum cure times (t_90_) were calculated from the curves of each material as the time required to reach 90% (S’_t90_) of the change from minimum torque toward maximum that can be calculated using Equation (1).
(1)$$S\prime_{t90}=0.9\;\left(S\prime_{max}-S\prime_{min}\right)+S\prime_{min}.$$

#### 2.2.12. Curing of Rubber Compounds

The rubber compounds were cured by compression molding at 150 °C and 150 bars to the calculated t_90_ values of each compounds. The curing was performed in a Labtech Engineering hot press and a stainless-steel mold with a rectangular geometry (80 × 30 × 2 mm) to which approximately 6 g of compound was added yielding a cured sheet with a thickness in the range of 1.8–2.0 mm.

#### 2.2.13. Tensile Test

Stress-strain tests were performed on an electro-mechanical testing machine INSTRON 5967 as per DIN 53504, using type S2 dumbbell specimens, that is, 75 mm length, and benchmark distance of 20 mm. The rate of grip separation was 200 mm·min^−1^. Young moduli were calculated from the slope at the beginning of the strain stress curves up to 1.5% of elongation.

#### 2.2.14. Crosslinking Density

Small square samples were cut from the cured sample sheet (0.5 × 0.8 cm) and immersed in toluene at room temperature to assess the degree of swelling as a function of time. In this case, the swelling degree can be directly related to the crosslinking density. Experiments were done three times. Specimens from each composition were kept completely immersed in 20 mL of toluene throughout the test. The solvent was changed every 24 h. At given times, the specimens were removed from the solvent, had their surface carefully dried, weighed and were placed back into the solvent. Each specimen was kept out of the solvent for less than 20 s. The specimens were weighed in air on a balance with accuracy 0.1 mg. Crosslinking densities were calculated using the Flory-Rehner equation (see in Section A.2 for calculation details).

#### 2.2.15. Atomic Force Microscopy

The samples were trimmed and surfaced with a LEICA EM UC6 cryo-ultramicrotome at −120 °C to produce a flat surface of the cross-section for Atomic Force Microscopy (AFM) analysis. Images of the topography and nanomechanical properties (modulus and loss tangent) of the samples were acquired using the Amplitude Modulation-Frequency Modulation (AM-FM) mode of the MFP-3D Infinity AFM (Asylum Research). All measurements were made under ambient conditions (room temperature and relative humidity of about 50%) and a standard cantilever holder for operation in air was used. Images of 20 × 20 μm^2^ areas were taken with a resolution of 256 × 256 pixels at a scan rate of 1.5 Hz. Cantilevers’ spring constants used in this study were about 30 N·m^−1^ (AC160TS-R3 model from Olympus). The first and second resonant frequencies for AC160TS-R3 cantilevers were about 300 kHz and 1.6 MHz, respectively. To ensure repulsive intermittent contact mode, the amplitude setpoint was chosen as A_setpoint_/A_0_ ~ 0.20 so that the phase is well fixed below 90°. A relative calibration method was applied to estimate the tip radius using a dedicated reference sample kit provided by Bruker (Model: PFQNM-SMPKIT-12m). The deflection sensitivity and the spring constant of the cantilever were determined using the GetReal™ Automated Probe Calibration feature. Using the polystyrene/low density polyethylene (PS/LDPE) standard sample, the tip radius was then adjusted to obtain the proper value of 2.7 GPa for the PS phase. The reported average and standard deviation values of modulus and loss tangent consider at least 5 and up to 10 images in each sample for reliable results.

### 2.3. Synthesis of Di-Benzoxazine Monomers

#### 2.3.1. 4,4′-Dihydroxydiphenyl Disulfide and Furfurylamine (4DPDS-fa)

The disulfide-containing benzoxazine (4DPDS-fa) was synthesized by Mannich condensation. 4DPDS (6 g., 24 mmoles, 0.5 eq.), furfurylamine (4.7 g., 48 mmoles, 1eq.) and paraformaldehyde (3 g., 96 mmoles, 2 eq.) were reacted in toluene in a round bottom flask under mechanical stirring at 110 °C for 5 h. After the reaction the solvent was evaporated under reduced pressure. Then, the product was solubilized in CHCl_3_ and three liquid–liquid extraction with 2 N NaOH and three with distilled water were carried out. The organic layer was dried over magnesium sulphate, then filtered and the solvent evaporated under reduced pressure. The final product was dried for 4 h under reduced pressure (< 1mBar) at 100 °C. Yield = 90%.

^1^H NMR (CDCl_3_, 600 MHz, 298 K), δ (ppm) = (assignment, [attribution], experimental integration, theoretical integration). δ = 3.89 (N-CH_2_, [d], exp 2.02 H, th 2.00 H); 3.97 (N-CH_2_-Ar, [2], exp 2.02 H, th 2.00 H); 4.88 (N-CH_2_-O, [1], integration reference 2.00 H); 6.24 (-CH=CH*-C-, [e], exp 1.08 H, th 1.00 H); 6.33 (-CH*=CH-C-, [f], exp 1.04 H, th 1.00 H); 6.75 (-CH-CH*=C-S-, [c], exp 1.06 H, th 1.00 H); 7.09 (-C=CH-C-S-, [a], exp 1.07 H, th 1.00 H); 7.24 (-CH*-CH=C-S-, [b], exp-H, th 1.00 H); 7.41 (-CH=CH*-O-, [g], exp 1.05 H, th 1.00 H)

^13^C NMR (CDCl_3_, 600 MHz, 298 K), δ (ppm) = (assignment, [attribution]). δ = 48.2 (N-CH_2_, [d]); 49.4 (N-CH_2_-Ar, [2]); 82.0 (N-CH_2_-O, [1]); 109.2 (-CH=CH*-C-, [e]); 110.3 (-CH*=CH-C-, [f]); 117.4 (-CH-CH*=C-S-, [c]); 120.3 (-CH_2_-C*-CH-, [i]); 128.5 (C-S, [j]); 130.2 (-C=CH*-C-S-, [a]); 130.6 (-CH*-CH=C-S-, [b]); 142.7 (-CH=CH*-O-, [g]); 151.2 (-CH=C*-O-, [k]); 154.3 (CH=C-O, [h]).

FTIR (cm^−1^): 1571 (stretching of furan ring), 1227 (C-O-C oxazine asymmetric stretching), 929 (out-of-plane bending vibration of the benzene ring).

Elemental analysis: element (exp, th); N (3.2, 3.5); C (44.3, 44.8); H (42.7, 41.4); S (3.4, 3.4); O (6.4, 6.9)

#### 2.3.2. 3,3′-Dihydroxydiphenyl Disulfide and Furfurylamine (3DPDS-fa)

The disulfide-containing benzoxazine (3DPDS-fa) was synthesized by Mannich condensation. 3DPDS (6.3 g., 25 mmoles, 0.5 eq.), furfurylamine (4.8 g., 50 mmoles, 1eq.) and paraformaldehyde (3.2 g., 100 mmoles, 2 eq.) were reacted in toluene in a round bottom flask under mechanical stirring at 110 °C for 5 h. After the reaction the solvent was evaporated under reduced pressure. Then, the product was solubilized in CHCl_3_ and three liquid-liquid extraction with 2 N NaOH and three with distilled water were carried out. The organic layer was dried over magnesium sulphate, then filtered and the solvent evaporated under reduced pressure. The final product was dried for 4 h under reduced pressure (<1mBar) at 100 °C. Yield = 91%.

^1^H NMR (CDCl_3_, 600 MHz, 298 K), δ (ppm) = (assignment, [attribution], experimental integration, theoretical integration). δ = 3.85, 3.89, 3.91 (N-CH_2_, [d’, d’, d], exp 2.02 H, th 2.00 H); 3.98, 4.07, 4.13 (N-CH_2_-Ar, [2, 2′, 2′], exp 2.09 H, th 2.00 H); 4.84, 4.86, 4.87 (N-CH_2_-O, [1′, 1′, 1], integration reference 2.00 H); 6.25 (-CH=CH*-C-, [e], exp 1.00 H, th 1.00 H); 6.33 (-CH*=CH-C-, [f], exp 1.01 H, th 1.00 H); 6.75–7.22 (aromatic protons, [a, b, c], exp 3.01 H, th 3.00 H); 7.41 (-CH=CH*-O-, [g], exp 1.00 H, th 1.00 H)

^13^C NMR (CDCl_3_, 600 MHz, 298 K), δ (ppm) = (assignment, [attribution]). δ = 48.3 (N-CH_2_, [d]); 48.2, 48.4, 49.4 (N-CH_2_-Ar, [2]); 81.3, 81.4, 82.0 (N-CH_2_-O, [1]); 109.2 (-CH=CH*-C-, [e]); 110.4 (-CH*=CH-C-, [f]); 118.9 (-CH_2_-C*-CH-, [i]); 115.4–128.4 (aromatic carbons, [a, b, c]); 136.6 (C-S, [j]); 142.8 (-CH=CH*-O-, [g]); 151.5 (-CH=C*-O-, [k]); 154.6 (CH=C-O, [h]).

FTIR (cm^−1^): 1569 (stretching of furan ring), 1220 (C-O-C oxazine asymmetric stretching), 936 (out-of-plane bending vibration of the benzene ring).

Elemental analysis: element (exp, th); N (3.2, 3.5); C (44.9, 44.8); H (42.0, 41.4); S (3.9, 3.5); O (6.0, 6.9)

#### 2.3.3. 4,4′-Thiodiphenol and Furfurylamine (4DTP-fa)

The sulfur-containing benzoxazine (4DTP-fa) was synthesized by Mannich condensation. 4DTP (6 g., 27.5 mmoles, 0.5 eq.), furfurylamine (5.3 g., 55 mmoles, 1eq.) and paraformaldehyde (3.5 g., 110 mmoles, 2 eq.) were reacted in toluene in a round bottom flask under mechanical stirring at 110 °C for 4 h. After the reaction the solvent was evaporated under reduced pressure. Then, the product was solubilized in CHCl_3_ and three liquid–liquid extraction with 2 N NaOH and three with distilled water were carried out. The organic layer was dried over magnesium sulphate, then filtered and the solvent evaporated under reduced pressure. The final product was dried for 4 h under reduced pressure (< 1mBar) at 100 °C. Yield = 88%.

^1^H NMR (CDCl_3_, 600 MHz, 298 K), δ (ppm) = (assignment, [attribution], experimental integration, theoretical integration). δ = 3.91 (N-CH_2_, [d], exp 2.09 H, th 2.00 H); 3.97 (N-CH_2_-Ar, [2], exp 2.01 H, th 2.00 H); 4.87 (N-CH_2_-O, [1], integration reference 2.00 H); 6.25 (-CH=CH*-C-, [e], exp 1.02 H, th 1.00 H); 6.33 (-CH*=CH-C-, [f], exp 1.03 H, th 1.00 H); 6.75 (-CH-CH*=C-S-, [c], exp 1.00 H, th 1.00 H); 6.97 (-C=CH-C-S-, [b], exp 0.99 H, th 1.00 H); 7.11 (-CH*-CH=C-S-, [a], exp 0.99 H, th 1.00 H); 7.41 (-CH=CH*-O-, [g], exp 1.00 H, th 1.00 H)

^13^C NMR (CDCl_3_, 600 MHz, 298 K), δ (ppm) = (assignment, [attribution]). δ = 48.6 (N-CH_2_, [d]); 49.4 (N-CH_2_-Ar, [2]); 82.0 (N-CH_2_-O, [1]); 109.1 (-CH=CH*-C-, [e]); 110.2 (-CH*=CH-C-, [f]); 117.5 (-CH-CH*=C-S-, [a]); 120.5 (-CH_2_-C*-CH-, [i]); 127.3 (C-S, [j]); 130.6 (-C=CH*-C-S-, [c]); 131.1 (-CH*-CH=C-S-, [b]); 142.7 (-CH=CH*-O-, [g]); 151.4 (-CH=C*-O-, [k]); 153.4 (CH=C-O, [h]).

FTIR (cm^−1^): 1573 (stretching of furan ring), 1226 (C-O-C oxazine asymmetric stretching), 930 (out-of-plane bending vibration of the benzene ring).

Elemental analysis: element (exp, th); N (3.2, 3.5); C (46.4, 45.6); H (42.2, 42.1); S (1.8, 1.8); O (6.3, 7.0)

#### 2.3.4. Bisphenol A and Furfurylamine (BA-fa)

The model benzoxazine from bisphenol A and furfurylamine (BA-fa) was synthesized by Mannich condensation following a procedure already reported [36]. Bisphenol A (6 g., 26 mmoles, 0.5 eq.), furfurylamine (5.1 g., 52 mmoles, 1eq.) and paraformaldehyde (3.1 g., 104 mmoles, 2 eq.) were reacted in toluene in a round bottom flask under mechanical stirring at 110 °C for 6 h. After the reaction the solvent was evaporated under reduced pressure. Then, the product was solubilized in CHCl_3_ and three liquid–liquid extraction with 2 N NaOH and three with distilled water were carried out. The organic layer was dried over magnesium sulphate, then filtered and the solvent evaporated under reduced pressure. The final product was dried for 4 h under reduced pressure (<1mBar) at 100 °C. Yield = 85%.

^1^H NMR (CDCl_3_, 600 MHz, 298 K), δ (ppm) = (assignment, [attribution], experimental integration, theoretical integration). δ = 1.59 (-CH_3_, [A], exp 4.18 H, th 4.00 H); 3.94 (N-CH_2_, [d], exp 2.07 H, th 2.00 H); 3.99 (N-CH_2_-Ar, [2], exp 2.01 H, th 2.00 H); 4.85 (N-CH_2_-O, [1], integration reference 2.00 H); 6.26 (-CH=CH*-C-, [e], exp 1.00 H, th 1.00 H); 6.33 (-CH*=CH-C-, [f], exp 1.03 H, th 1.00 H); 6.71 (-O-C=CH*-CH, [c], exp 1.00 H, th 1.00 H); 6.81 (-C=CH-C-, [a], exp 0.99 H, th 1.00 H); 7.11 (-O-C=CH-CH*, [b], exp 1.01 H, th 1.00 H); 7.41 (-CH=CH*-O-, [g], exp 1.00 H, th 1.00 H)

^13^C NMR (CDCl_3_, 600 MHz, 298 K), δ (ppm) = (assignment, [attribution]). δ = 31.2 (-CH_3_, [A]); 41.9 (CH_3_-C*-C-, [l]); 48.5 (N-CH_2_, [d]); 50.1 (N-CH_2_-Ar, [2]); 81.8 (N-CH_2_-O, [1]); 109.1 (-CH=CH*-C-, [e]); 110.3 (-CH*=CH-C-, [f]); 116.1 (-O-C=CH*-CH, [c]); 118.9 (-CH_2_-C*-CH-, [i]); 125.5 (-C=CH-C-, [a]); 126.5 (-O-C=CH-CH*, [b]); 142.7 (-CH=CH*-O-, [g]); 143.4 (CH_3_-C-C*-, [j]); 151.8 (-CH=C*-O-, [k]); 151.9 (CH=C-O, [h]).

FTIR (cm^−1^): 1586 (stretching of furan ring), 1230 (C-O-C oxazine asymmetric stretching), 936 (out-of-plane bending vibration of the benzene ring).

Elemental analysis: element (exp, th); N (2.6, 3.1); C (44.4, 44.6); H (47.4, 46.2); O (5.6, 6.1)

## 3. Results and Discussion

### 3.1. Synthesis and Molecular Characterization of Benzoxazines

For the purpose of this work, two new dibenzoxazine (diBz) monomers containing two consecutive heteroatoms of sulfur were designed (Scheme 2). First, 4,4′-dihydroxydiphenyl disulfide (4DPDS) was reacted with furfurylamine (fa) in the presence of paraformaldehyde to form 4DPDS-fa (Scheme 2a). Secondly, 3,3’-dihydroxydiphenyl disulfide (3DPDS), which differs from 4DPDS by the relative position of the hydroxyl group (O‒H) regarding the disulfide bond (S‒S), was employed as a synthon for the design of 3DPDS-fa (Scheme 2b). Furfurylamine was used for its bio-based origin and its known involvement in the benzoxazine network formation, resulting in increased its glass transition temperature (T_g_) [36,37,38].

The structural features of 4DPDS-fa, and 3DPDS-fa were characterized by ^1^H NMR, ^13^C NMR, FTIR, and Raman (see in Appendix B Figures from Figure A1, Figure A2, Figure A3, Figure A4, Figure A5, Figure A6 and Figure A7) and the purity was verified by the elemental analysis. 

The formation of the Bz monomers is revealed by the presence of peaks at 4.8 ppm and 3.9 ppm which correspond to O‒CH_2_‒N [1] and Ar‒CH_2_‒N [2] respectively (Figure 1). The NMR spectrum of 3DPDS-fa features more peaks than its congeners. Indeed, the reaction of 3DPDS with fa led to a mixture of three isomers as depicted in Scheme 2 (b_1_, b_2_, and b_3_). This is proven by the appearance of additional peaks next to the common peaks at 4.86 and 4.84 ppm, at 4.13 and 4.07, and at 3.88 and 3.85 ppm (marked with * in Figure 1b). Moreover, one singlet and two doublets are observed in the aromatic region for 4DPDS-fa as expected. Nevertheless, in the case of 3DPDS-fa, new aromatic peaks appear attesting of the formation of more than one diBz structure.

Additionally, the success of the synthesis was confirmed by FTIR with the appearance of new peaks around 1225 cm^−1^ and 930 cm^−1^ which corresponds to C‒O‒C oxazine asymmetric stretching and out-of-plane bending vibration of the benzene ring respectively (see in Appendix B
Figure A5 and Figure A6). 

Furthermore, the presence of the disulfide in 3DPDS-fa and 4DPDS-fa was confirmed by Raman spectroscopy (see in Appendix B
Figure A7). It is worth indicating that around 2% of the S‒S bonds of 4DPDS-fa cleaved, as shown by the small peak located between 2550 and 2600 cm^−1^, which is attributed to thiol (S‒H) bonds. The reduction of the disulfide bonds into thiols would presumably be due to the primary amine group of furfurylamine. As recently put into the spotlight, a benzoxazine containing disulfide bonds might cleave and rearrange upon heating [39]. The presence of these thiols indicates the disulfide bond of 4DPDS-fa is also able to cleave in relatively mild conditions. This could be considered as an asset since they could accelerate the ROP of the benzoxazine precursor due to their known catalytic effect [40,41]. On the contrary, no S‒H contributions were observed for 3DPDS-fa. The apparent higher stability of 3DPDS-fa compared to 4DPDS-fa is difficult to justify without resorting to molecular dynamics. Nevertheless, the bond dissociation energy (BDE) of disulfide bonds is known to be strongly influenced by the chemical and electronic environments. Significant variations of the BDE of 4,4′ and 2,2′-dihydroxydiphenyl disulfide have already been reported [42]. To the best of our knowledge, the BDE of 3,3′-dihydroxydiphenyl disulfide has never been reported, but it is reasonable to expect it would differ from the other isomers. This, together with additional facts given all along this manuscript, tends to converge to the conclusion that the disulfide bond of 3DPDS-fa is more stable than in 4DPDS-fa.

### 3.2. Thermal Properties and Curing Behavior of Benzoxazine Monomers

The thermal properties of the novel synthesized dibenzoxazines were studied by DSC (Figure 2). An endothermic peak corresponding to the melting temperature is observed for each precursor. For 4DPDS-fa, this peak is centered at 74 °C. In the case of 3DPDS-fa, the endothermic peak presents two main contributions, with a first peak at 90 °C and a second at 115 °C. As 3DPDS-fa is a mixture of isomers, it makes sense that more than one melting peak is observed by DSC. An exothermic peak is observed for each Bz, corresponding to the thermally activated ring opening polymerization (ROP). 3DPDS-fa has a single exothermic peak centered at 214 °C (Figure 2b), while 4DPDS-fa exhibits two exothermic peaks (Figure 2a). The exothermic peak located at the lowest temperature (i.e., 215 °C) is attributable to the benzoxazine ring opening. The peak at higher temperatures, centered at 240 °C is attributed to the post-polymerization reaction of furan rings, as previously reported [36,37,38]. 

The involvement of the furan ring in the polymerization was double-checked by FTIR (see in Appendix C
Figure A8) showing a widening around 1570–1580 cm^−1^, indicating that electrophilic substitutions occurred on the furfuryl groups of 4DPDS-fa. Even though the DSC profile of 3DPDS-fa just reveals one exothermic peak, the furan rings were also involved in the polymerization process as attested by FTIR. An increase of the absorption peak located at 1570 cm^−1^ gives the evidence the reaction occurred (see in Appendix C
Figure A9). Interestingly, it seems the involvement of the furan ring during the benzoxazine curing process occurs at lower temperatures. Finally, the enthalpies of polymerization for 4DPDS-fa and 3DPDS-fa are similar, slightly lower for 4DPDS-fa (Table 2, columns 4 and 5). 

It is worth indicating that each monomer does not release volatile species at least up to 220 °C as attested by TGA measurements. It is not below 264 °C that 3DPDS-fa releases 5% degradation products when heated. For 4DPDS-fa, the temperature of 5% of weight loss (T_5%_) is found to be 250 °C. (Table 2, column 6; Figure A10 in Appendix C).

The evolution of the complex viscosity (η*) of each Bz heated under isothermal conditions at 150 °C is displayed in Figure 3. In the molten state, the monomers exhibit very low values of η*, between 6·10^2^ and 8·10^2^ mPa·s. After a few minutes, η* noticeably increases, indicating the curing of the monomers. The gelation times (t_gel_), defined as the crossover point of G’ and G’’, are observed at 17 and 41 min, for 4DPDS-fa and 3DPDS-fa, respectively (see in Appendix C
Figure A11). Surprisingly, 3DPDS-fa takes more than twice the time to cure in comparison to 4DPDS-fa. One of the possible explanations for this significant difference in the curing behaviors of 3DPDS-fa and 4DPDS-fa may be attributed to the small amount of thiol groups present in the 4DPDS-fa compound identified by Raman spectroscopy. Indeed, thiols are known to catalyze the ring opening polymerization of benzoxazine [40,41]. 

To further investigate the curing behavior of 4DPDS-fa, a third benzoxazine monomer was synthesized from 4,4′-thiodiphenol and furfurylamine (4DTP-fa). In this molecule the disulfide bond was replaced by a single sulfur heteroatom (Scheme 2c). 4DTP-fa is not able to form thiols while it keeps the same chemical environment than 4DPDS-fa (Scheme 3a,c). All the molecular and thermal characterizations of 4DTP-fa are reported in the Section C.1) (Figures from Figure A12, Figure A13, Figure A14, Figure A15, Figure A16, Figure A17 and Figure A18), and the detail of its thermal behavior is reported in Table 2, row 4. The curing behavior of 4DTP-fa is reported in Figure 3 to be compared to the profile of 3DPDS-fa and 4DPDS-fa. 4DPT-fa behaves similarly to 3DPDS-fa, reaching the gelation time after 34 min. This result supports the assumption of a catalytic effect triggered by the presence of thiols in 4DPDS-fa. 

Thereafter, each benzoxazine precursor was cured following the procedure described in the experimental part and shaped to analyze their glass transition temperature (T_g_) (see the Section C.2 Figures from Figure A19, Figure A20 and Figure A21). In all cases, the T_g_ was measured to be above 250 °C (Table 2, column 8) showing the potential of these diBz to act as reinforcing resins. Indeed, they exhibit higher T_g_ than currently used phenolic resins (i.e., ~170 °C) [14]. These values together with the reactivity of the monomers make them suitable candidates as reinforcing resins for rubber applications.

### 3.3. Investigation of Benzoxazine Curing in the Presence of Sulfur

Ishida et al. demonstrated that elemental sulfur is an efficient catalyst to decrease the polymerization temperature of benzoxazine due to the formation of *in-situ* initiators [35]. This triggering effect was reported to happen at temperatures below the thermally induced homolytic cleavage of sulfur (Scheme 4a). Additionally, Liu et al. reported a reaction occurring between sulfur and benzoxazine to form a copolymer at temperatures above 159 °C (Scheme 4b) [30]. This reaction is initiated by sulfur radicals formed after the cleavage of sulfur ring and it is known as sulfur radical transfer and coupling reaction (SRTC). In the following paragraphs, the effect of sulfur on the polymerization of 4DPDS‑fa, 3DPDS-fa, and 4DTP-fa is discussed. 

For this purpose, each benzoxazine was mixed with sulfur at different amounts, and their thermograms were recorded by DSC (see in Section C.3 Figures from Figure A22, Figure A23 and Figure A24). For all the tested benzoxazines, when sulfur is added, the maximum temperature of the exothermic peak (T_p,max_) is reduced (Figure 4a). In the case of 3DPDS-fa the maximum of the exotherm is greatly shifted from 214 to 177 °C. On the other hand, the exothermic peak of 4DPDS-fa was converted from a double wide peak located around 215 °C to a single narrow peak centered at 187 °C. Similar results were observed for 4DTP-fa with a decrease of T_p,max_ of about 36 °C. However, for this monomer, the double wide peak around 224 °C was slowly converted to a single peak with the increase of the amount of sulfur (Figure A24 in Section C.3). A single narrow peak centered at 188 °C was obtained after the addition of 20 wt % of sulfur. Furthermore, the dependency between the amount of sulfur and the T_p,max_ is displayed in Figure 4b. The differences observed for 4DTP-fa in the tendency are explained by the gradual catalytic effect of sulfur on the double exothermic peak. Nevertheless, the higher the amount of sulfur, the lower the polymerization temperature for all benzoxazines precursors. These results are in agreement with previously reported work [34,35]. 

Rheo-kinetic analyses were also carried out to detail the catalytic effect of sulfur on the monomers from a viscoelastic perspective. For this purpose, each benzoxazine precursor was mixed with different amounts of sulfur and was analyzed under isothermal conditions at 150 °C (see Figures from Figure A25, Figure A26 and Figure A27 in Section C.3). The curves obtained for 3DPDS-fa, 4DPDS-fa, and 4DTP-fa without and with 5 wt % of sulfur are displayed in Figure 5a. Figure 5b depicts the evolution of the gelation times of each benzoxazine precursor as a function of the amount of sulfur added.

As expected, the polymerization of 3DPDS-fa is strongly catalyzed by sulfur, even if it is added in small amounts. When 1 or 5 wt % of sulfur is employed, the gelation time is shifted from 41 min to 17 or 9 min, respectively. On the contrary, the polymerization of 4DPDS-fa is almost not affected by the presence of sulfur. As previously mentioned, this might be due to the small amount of thiolated forms of cleaved 4DPDS-fa, which already catalyze its polymerization. This assumption is also confirmed by the curing behavior of 4DTP-fa in the presence of sulfur as a high catalytic effect is also observed. This mono-sulfide benzoxazine precursor exhibits a shift of the t_gel_ from 34 min to 15 min with 5 wt % of sulfur. 

Finally, each precursor was polymerized in the absence and presence of sulfur following the procedure described in the experimental part and their solubility was tested in dimethylformamide (DMF). For the sake of clarity, cured benzoxazine precursors will be annotated as poly (“precursor acronym”). When cured in the presence of sulfur, they will be annotated poly (“precursor acronym”)/S_8_. Poly(3DPDS-fa), poly(4DPDS-fa), and poly(4DTP-fa) were insoluble, as expected from polybenzoxazine networks. On the contrary, poly(4DPDS-fa)/S_8_ was fully soluble in DMF even when it was cured in the same conditions. A partial solubility was also observed for poly(3DPDS-fa)/S_8_ and poly(4DTP-fa)/S_8_, for which a solid fraction of about 60% of the initial mass was recovered after the test. The partial solubility of poly(3DPDS-fa)/S_8_ and poly(4DTP-fa)/S_8_ could be explained by a co-reaction occurring between sulfur and the oxazine rings in these specific conditions (Scheme 4b) [30]. The full solubility of poly(4DPDS-fa) in DMF indicates an additional reaction is occurring between S_8_ and 4DPDS-fa otherwise its solubility should remain the same than poly(4DTP-fa)/S_8_. The only structural difference between these molecules is the disulfide bond. Therefore, the other side-reaction taking place could be through this bond as reported in other works and described in Scheme 3b, with the formation of a polysulfide [2,43,44]. 

Interestingly, the disulfide bond in 3DPDS-fa does not undergo a similar side-reaction. Indeed, poly(3DPDS-fa)/S_8_ shows a similar solubility than poly(4DTP-fa)/S_8_, meaning their curing in the presence of sulfur are also similar. The two different curing behaviors between poly(3DPDS-fa)/S_8_ and poly(4DPDS-fa)/S_8_ may be explained by the lower stability of the disulfide bond of 4DPDS-fa, as above-mentioned. 

In conclusion, three novel benzoxazines containing sulfur have been synthesized. Due to their curing kinetic in the presence or absence of sulfur and their high T_g_ above 250 °C, 4DPDS-fa, 3DPDS-fa, and 4DTP-fa are three suitable candidates to be used for the reinforcement of rubber compounds. The performance of each precursor as well as the impact of the disulfide stability on the curing behavior and the final properties of the compounds will be reported in the following section. 

### 3.4. Application in Rubber Compounds

The potential of each benzoxazine to act as a reinforcing resin in rubber compounds has been tested in a typical recipe composed of polyisoprene and a curing system constituted of sulfur, N,N′-dicyclohexyl benzothiazole-2-sulphenamide (DCBS) as accelerator, zinc oxide (ZnO) and stearic acid (SA). For the sake of clarity, the term “compound” will be employed in the rest of the manuscript to designate mixture of polyisoprene and curing system with or without benzoxazine precursors.

It is worth noting that the curing kinetics of benzoxazine precursors have also been assessed in the presence of the ingredients of the curing package used to crosslink polyisoprene. These thermal characterizations can be found in Appendix D (Figure A28). They reveal that stearic acid also affects the curing of the benzoxazine monomers, but to a significantly lower extent compared to sulfur. 

Rubber compounds were mixed in an internal mixer in a two-step process, described in detail in the experimental part. It should be noted that the benzoxazines mixed with the rubber are introduced as precursors. Their curing is occurring simultaneously with the curing of the rubber. All the experiments described in the section above demonstrated that each benzoxazine precursor would be able to crosslink in the same time than the rubber during the curing step. The benzoxazine precursors were added to the polyisoprene formulations at different loading. For the sake of clarity and to avoid any confusion, polyisoprene compounds containing a benzoxazine are annotated as PI(benzoxazine)_phr_. For instance, a polyisoprene compound containing 10 phr of 4DPDS-fa is called PI(4DPDS-fa)_10_. 

#### 3.4.1. Curing of Rubber Compounds Containing Benzoxazine Precursors

The curing process of the rubber compounds containing the benzoxazine precursors was followed by using a moving die rheometer (MDR). In this method, the curing of the rubber is followed thanks to the evolution of the torque. The torque increases as the material is crosslinking, being more resistant to the torsional strength applied. The optimum cure time (t_90_) is then determined following Equation (1). It is worth indicating that this time is a reference parameter in the rubber industry. The t_90_ values and the maximum achieved torques (S’_max_) obtained for each compound are gathered in Table 3, columns 2 and 3, respectively. The curing profiles of the rubber compounds are depicted in Figure A29 in the Appendix D. 

In the case of raw polyisoprene compound (PI), so-called reference, t_90_ was reached after 27 min at 150 °C. PI(3DPDS-fa) cured in a comparable duration than the reference ranging between 26 and 29 min, depending on the amount of resin. In the case of PI(4DPDS-fa), the curing time was strongly affected, t_90_ being reached after only 16 min when 15 phr of the resin were employed. In this case, the t_90_ decrease was also accompanied by a significant drop of the maximum achievable torque, up to 73% (Table 3, rows from 6 to 8). It is worth indicating that the crosslinking density of these rubber compounds were also drastically decreased in comparison to the reference (Table 3, column 4). Indeed, the values of crosslinking density (ν_c_) decreased to 0.3·10^−4^ mol·cm^−3^, compared to ~2.0·10^−4^ mol·cm^−3^ for the reference and PI(3DPDS-fa). 

The decrease of the maximum achievable torque together with the drop of the crosslinking density when 4DPDS-fa is used can be explained by the lower stability of the disulfide bond, as previously discussed. When a disulfide bond breaks, sulfur radicals are formed. One of the possibilities is that these radicals form thiols as observed by Raman (Scheme 3a). Another option is that they could trap other sulfur radicals coming from the opening of the elemental sulfur ring (S_8_). Therefore, polysulfide chains will be formed in between the phenolic moieties as previously reported (Scheme 3b) [2,43,44]. If such competitive reactions occur during the curing of PI(4DPDS-fa), less sulfur would be accessible for the curing of polyisoprene and thus, the properties of the compounds would be affected. This hypothesis is supported by the maximum torque reduction observed by MDR as well as the drop of the crosslinking density for PI(4DPDS-fa). Additionally, the solubility test of poly(4DPDS-fa) with and without sulfur described above revealed that a fully soluble network was formed in the presence of sulfur. These results are a further proof that a reaction is occurring between 4DPDS-fa and sulfur. On the contrary, when 3DPDS-fa was employed, the values of S’_max_ and ν_c_ remained similar to the reference, emphasizing the absence of a similar reaction between the benzoxazine precursor and S_8_.

It is also worthy to mention that S’_max_ and ν_c_ of polyisoprene compounds containing 4DTP-fa are in the same range than the reference. It indicates that the curing of this mono-heteroatom benzoxazine precursor, simultaneously to polyisoprene, does not consume sulfur and, if it happens, it does not affect the curing of the rubber network. This provides additional evidence that the low stability of the S‒S bond in 4DPDS-fa is the reason for the low curing extent of its compound with polyisoprene and the low crosslinking density. 

#### 3.4.2. Morphological Characterization and Nanomechanical Properties of Cured Rubber Compounds by AFM

The morphology of the compounds containing 15 phr of benzoxazine was explored by AFM. The mapping of the mechanical properties of cured polyisoprene compounds with and without benzoxazine precursors in areas of 20 × 20 μm^2^ are depicted in Figure 6a. The morphologies of the compounds are revealed by the contrast in nanomechanical properties. In the reference compound, small stiff irregular particles were observed (white contrast), probably related to zinc oxide particles present in the composition. In the compounds containing benzoxazine, round shape domains with high modulus can be seen well dispersed in the matrix. Despite a high dispersity of nodules sizes, there is a clear trend where 4DPDS-fa nodules are almost undetectable as shown in Figure 6a. This morphology is aligned with the previous observations. Indeed, 4DPDS-fa is cleaving through the S‒S bond, with a polysulfide chain growing between the phenolic moieties, as illustrated on Scheme 3b. This cleavage leads to a dilution of the resin within the polyisoprene matrix. In these conditions, the scattered benzoxazine groups from 4DPDS-fa have a lower probability to meet with each other and thus, to form a network, in comparison to 3DPDS-fa and 4DTP-fa. This hypothesis is supported by the small size of polyBz domains found in PI(4DPDS-fa)_15_. 

The evaluation of the quantitative nanomechanical measurements acquired with the images follows the same conclusions. The average modulus for the rubber and polybenzoxazine phases in each sample are plotted in Figure 6b. The modulus of the rubber matrix in the reference sample was measured as 1.17 ± 0.16 GPa. It is noteworthy that the apparent high modulus of the rubber matrix is related to the high frequency used in the analysis (~1.6 MHz), since the viscoelastic properties of the rubber are frequency dependent. Nevertheless, a clear trend can be observed when considering the compounds containing benzoxazines as the modulus of the polyisoprene phase (E_PI_) is decreasing as follows: 4DTP-fa > 3DPDS-fa > 4DPDS-fa. In PI(4DTP-fa)_15_ and PI(3DPDS-fa)_15_, E_PI_s are similar than the E_PI_ values for neat crosslinked polyisoprene, in agreement with the crosslinking density measurements reported in the section above. The quantitative nanomechanical measurements are also in line with the drastic reduction of the crosslinking density for PI(4DPDS-fa)_15_, since E_PI_ significantly decreased by 43%. AM-FM also allows the simultaneous measurement of the loss tangent of each phase, complementarily to the modulus measurements. Results showed equivalent trends and are in accordance with the modulus measurements (see Figure A30 in Section D.1). These measurements, together with the lower crosslinking density, are well aligned with the hypothesis of a consumption of sulfur as stated previously due to a reaction with the 4DPDS-fa disulfide bond. 

Finally, the moduli of the polybenzoxazine nodules (E^PI^_polyBz_) within the compounds were measured and compared to the moduli of neat polyBz (E_polyBz_) (see Figure A31 in Section D.1). In PI(3DPDS-fa)_15_ and PI(4DTP-fa)_15_, E^PI^_polyBz_ are slightly lower than E_polyBz_, but they remain in the same range (around 2.1 GPa instead of 3 GPa). These values prove the successful crosslinking of the benzoxazine precursors simultaneously with the rubber curing. On the contrary, for PI(4DPDS-fa)_15_, moduli of both rubber and polybenzoxazine phases are lower than their corresponding references (0.67 and 1.4 GPa for E_PI_ and E^PI^_polyBz_ respectively, compared to 1.17 and 3 GPa for neat rubber and polyBz references). This also agrees with the assumption that sulfur is consumed by this benzoxazine, this last one being diluted and unable to efficiently crosslink and preventing at the same time a proper curing of polyisoprene. 

#### 3.4.3. Tensile Test of Cured Rubber Compounds 

Tensile tests were performed to assess the mechanical properties of the vulcanized compounds of polyisoprene and benzoxazine precursor (see Figure A32 in Section D.2). A summary of these properties is gathered in Table 4. 

The cured compound PI(4DPDS-fa) exhibits poor mechanical properties, below the reference for all the compositions tested (Figure 7). This result was expected because of the low curing extent of the compound, shown by the low crosslinking density and low E_PI_ and E^PI^_polyBz_ measured by AFM. For PI(3DPDS-fa) and PI(4DTP-fa), a reinforcement of the mechanical properties was observed as attested by the highest values of Young modulus (up to 3.1 and 2.9 MPa respectively) compared to the reference (2.6 MPa). Furthermore, the stresses measured at 200% of strain also increased for these two compounds (Figure 7). It is important to highlight that the ultimate mechanical properties of PI(3DPDS-fa) were also improved compared to the reference, with a higher elongation and stress at break up to ~700% and 15.1 MPa respectively, compared to ~500% and 11.3 MPa for the reference. From an overall perspective, PI(3DPDS-fa)_15_ is the compound with the most significant improvement of the stress at low (σ_200% of strain_) and high (σ_600% of strain_) elongation as well as enhancing the tensile strength of the compound.

For the sake of comparison, a model benzoxazine from bisphenol A and furfurylamine (BA-fa) was synthesized following a procedure previously reported [36]. The synthesis and characterization of this model molecule is given in the experimental part and the molecular characterization can be found in the Section D.3 (Figures from Figure A33, Figure A34 and Figure A35). BA-fa was then tested in the rubber formulation (PI(BA-fa)). Stress-strain curves are displayed in Figure A36 in Section D.3 and the mechanical properties are gathered in Table A2. This compound exhibits higher values of Young modulus than the reference demonstrating a reinforcement with this molecule is also feasible. However, the reinforcement observed in PI(BA-fa) is lower than for PI(3DPDS-fa) and PI(4DTP-fa), whatever the content of benzoxazine precursor. 

Finally, in order to evaluate the reinforcement effect of polybenzoxazine in comparison to phenolic resins, the mechanical properties of a compound containing phenolic resin (PR) were assessed. For that purpose, a phenolic system composed of a pre-condensed novolac resin and hexamethylenetetramine, as *in-situ* crosslinker, was mixed with the polyisoprene formulation (see more details about the formulation in Section D.4). Mechanical properties of phenolic resin compounds (PI(PR)_15_) were compared to the compound containing 3DPDS-fa (PI(3DPDS-fa)_15_) and the results are displayed in Figure 8 (see stress-strain curve in Section D.4, Figure A37).

Comparable results were obtained regarding Young’s modulus and stress at low strain (σ_200% of strain_). However, PI(3DPDS-fa)_15_ exhibited better stress at high strain (σ_600% of strain_). The ultimate tensile properties were also improved with the use of 3DPDS-fa showing higher elongation and stress at break than PI(PR)_15_ and PI reference. These results prove the feasibility to employ benzoxazines precursors to replace phenolic resins as reinforcing agents for rubber applications. 

## 4. Conclusions

Three new dibenzoxazines containing one (4DTP-fa) or two heteroatoms of sulfur (3DPDS-fa and 4DPDS-fa) were synthesized following a Mannich condensation reaction. The structural features of each benzoxazine precursor were characterized by ^1^H and ^13^C NMR, FTIR, and Raman. They revealed that about 2% of the disulfide bonds of 4DPDS-fa were cleaved to form thiol groups at the end of the synthesis, while in the case of 3DPDS-fa the disulfide bond remained intact. The structural differences between the two isomers appeared to support the stability of the disulfide bridge. As 4DTP-fa is composed of one atom of sulfur, it does not form thiols.

As shown by DSC and rheological measurements, the presence of thiols in 4DPDS-fa resulted in a significant catalytic effect of the curing when compared to 4DTP-fa and 3DPDS-fa. Indeed, isothermal curing at 150 °C revealed that the gelation point of 4DPDS-fa was reached only after 17 min, while it took 37 and 41 min for 4DTP-fa and 3DPDS-fa respectively. When the curing of each benzoxazine precursor was monitored in the presence of sulfur, a well-known catalyst for the ROP of benzoxazines, a strong catalytic effect was observed for 3DPDS-fa and 4DTP-fa, the gelation points being reached after 9 and 15 min respectively. On the contrary, the curing of 4DPDS-fa remained the same, confirming the catalysis by thiols. 

Thanks to their short gelation times at a relatively low temperature (150 °C), the curing of the three benzoxazine precursors fit the specifications for the curing of rubber compounds, and their suitability to be used as potential reinforcing resins in polyisoprene compound (PI) was tested. Freshly synthesized benzoxazine precursors were compounded with polyisoprene and curing additives containing sulfur. The curing of the compounds, so called PI (benzoxazine), brought substantial clues about the concomitant curing of PI and the benzoxazine monomers. On the one hand, 3DPDS-fa and 4DTP-fa did not interfere with the curing of polyisoprene. Indeed, similar t_90_ and crosslinking densities were measured for the compounds cured in the presence or absence of these benzoxazines. On the contrary, the compounds prepared with 4DPDS-fa, so-called PI(4DPDS-fa) did not reach the same extent of curing compared to the PI reference as shown by the lower values of t_90_ and crosslinking density together with the decrease of the maximum achieved torque. It was assumed that the low stability of the disulfide bond in 4DPDS-fa yielded a side-reaction when heated in the presence of sulfur, driving the consumption of this essential element for the crosslinking of polyisoprene. 

The morphology and nanomechanical properties of the compounds were explored by AFM and were in agreement with the previous results. In PI(3DPDS-fa) and PI(4DTP-fa) benzoxazine nodules of about 550 nm were observed and the moduli of each phase was similar to their respective reference. Conversely, AFM images of PI(4DPDS-fa) exhibited nodules of smaller sizes (around 440 nm). The moduli of both benzoxazine and rubber phases were also significantly lower than their corresponding references (0.67 and 1.4 GPa for E_PI_ and E^PI^_polyBz_ respectively, compared to 1.17 and 3 GPa for net rubber and polyBz references). This supports the assumption that sulfur is consumed by 4DPDS-fa, preventing an efficient curing of polyisoprene and leading to a detrimental impact on the polybenzoxazine network. 

Finally, results from the tensile tests of the cured compounds confirmed the previous observations. PI(3DPDS-fa)_15_ was the compound where the most significant improvement of the mechanical properties was observed. Furthermore, the reinforcement of the mechanical properties of the compound when using 3DPDS-fa was shown to be more significant than compounds prepared with a model benzoxazine or a commonly employed phenolic resin. These results confirmed the feasibility to employ benzoxazines precursors to reinforce rubber compounds and likewise replace phenolic resins for rubber applications. These investigations also provide a first guide to reinforce rubber compounds with benzoxazines. They highlight the importance of considering the curing kinetics of the benzoxazine precursors in the presence of the rubber curing additives, in particular sulfur.

## Data Availability

Data is contained within the article or supplementary material.

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
