# Peer review of "Synthesis of Novel Benzoxazines Containing Sulfur and Their Application in Rubber Compounds"

_polymers, 2021, doi:10.3390/polym13081262_

Round 1
Reviewer 1 Report
Thank you for your paper.
I would only suggest to rephrase R404 'each monomer remains stable' into something like 'no volatile species evolve' -> monomers are reactive by definition
My consideration is that the paper is ready for publications. I only list here some hints for the authors, which by the way did already an excellent job
Extraction of non reacted species: it would be significant to try solvent extraction (for instance by Soxhlet) of swollen cured rubber specimen of eventual non-reticulated benzoxazine monomers, to further confirm that crosslinking took place in the rubber environment at least for 4DTPfa and 3DPTSfa. The other compound may have low molecular weight extractables of 4DTPSfa combined with sulfur (non necessarily extractables, though)
An exact potential application field was not indicated by the authors. If rubber compounds for tires would be an option, then the compound study would need a filled system with carbon black, so that the properties at break of the reference compound would be more challenging to improve (higher than 20 MPa). Of course the AFM study would not be possible then.
Tensile tests: Also bad results deserve comments! What about Tensile strength of compounds with 5 phr of resins? For instance one may argue that low viscosity would help aggregation of resin in few big spots, acting as crack initiator
Author Response
Authors would like to thank the reviewers for their constructive comments and questions that will definitively increase the quality of the manuscript. We considered very attentively all the comments and a point-by-point answer is given below.
Reviewer #1
Thank you for your paper.
I would only suggest to rephrase R404 'each monomer remains stable' into something like 'no volatile species evolve' -> monomers are reactive by definition
My consideration is that the paper is ready for publications. I only list here some hints for the authors, which by the way did already an excellent job
We warmly thank the Reviewer for his nice comment. We are glad of such appreciation!
As suggested, we have modified this sentence accordingly: “It is worthy to indicate that each monomer does not release volatile species at least up to… “
Extraction of non reacted species: it would be significant to try solvent extraction (for instance by Soxhlet) of swollen cured rubber specimen of eventual non-reticulated benzoxazine monomers, to further confirm that crosslinking took place in the rubber environment at least for 4DTPfa and 3DPTSfa. The other compound may have low molecular weight extractables of 4DTPSfa combined with sulfur (non necessarily extractables, though)
We agree with the kind suggestion of the Reviewer. Actually, we tried but the characterization of the extracted part is really difficult due to the numerous extracted chemicals. The contribution of each benzoxazine was difficult to find in the spectra, and too low to be quantitative and comparative,
An exact potential application field was not indicated by the authors. If rubber compounds for tires would be an option, then the compound study would need a filled system with carbon black, so that the properties at break of the reference compound would be more challenging to improve (higher than 20 MPa). Of course the AFM study would not be possible then.
We fully agree with the Reviewer. Actually, this document is the “part A” of our work, the part B consisting in the study of carbon back filled compounds. We have collected so many data for the filled compounds what we decided to split in two documents, otherwise it would have been difficult to publish it due to the length.
Tensile tests: Also bad results deserve comments! What about Tensile strength of compounds with 5 phr of resins? For instance one may argue that low viscosity would help aggregation of resin in few big spots, acting as crack initiator
That’s right. We really appreciate this comment of the Reviewer. We focused more our comments on loading of 15 phr as it leads to the most significant results.

Reviewer 2 Report
The paper is aimed at a narrow groups of specialists. Many interesting results have been presented. The measurement methods are described very generally, but this is due to the fact that they are presented in the form of an article that cannot be too extensive.
Author Response
We would like to thank the Reviewer for his nice comment and appreciation of our work.